# Stiffness-Based Cell Setup Optimization for Robotic Deburring with a Rotary Table

Janez Gotlih *, Miran Brezocnik and Timi Karner

Faculty of Mechanical Engineering, University of Maribor, Smetanova ulica 17, 2000 Maribor, Slovenia; miran.brezocnik@um.si (M.B.); timi.karner@um.si (T.K.)
* Correspondence: janez.gotlih@um.si; Tel.: +386-2-220-7605

**Featured Application:** The proposed robotic deburring cell with integrated rotary table is designed to improve the performance of robotic deburring when deburring large workpieces.

**Abstract:** Deburring is recognized as an ideal technology for robotic automation. However, since the low stiffness of the robot can affect the deburring quality and the performance of an industrial robot is generally inhomogeneous over its workspace, a cell setup must be found that allows the robot to track the toolpath with the desired performance. In this work, the problems of robotic deburring are addressed by integrating components commonly used in the machining industry. A rotary table is integrated with the robotic deburring cell to increase the effective reach of the robot and enable it to machine a large workpiece. A genetic algorithm (GA) is used to optimize the placement of the workpiece based on the stiffness of the robot, and a local minimizer is used to maximize the stiffness of the robot along the deburring toolpath. During cutting motions, small table rotations are allowed so that the robot maintains high stiffness, and during non-cutting motions, large table rotations are allowed to reposition the workpiece. The stiffness of the robot is modeled by an artificial neural network (ANN). The results confirm the need to optimize the cell setup, since many optimizers cannot track the toolpath, while for the successful optimizers, a performance imbalance occurs along the toolpath.

**Keywords:** deburring; robot; stiffness; artificial neural network; genetic algorithm





## 1. Introduction

The increasing heterogeneity of industrial robots and the development of new manufacturing techniques have helped to extend the use of robots to a variety of non-optimized manufacturing processes. Nowadays, robots are more commonly used in machining than ever before. Some of the advantages of machining with robots, such as their price competitiveness, wide range of applications and flexibility, have proven to be beneficial for profitability compared to the general disadvantages, such as lower accuracies due to the low stiffness of the robot and higher cycle times when machining parts with long toolpaths [1]. Deburring is recognized as a technology where robots have a performance advantage over traditional methods such as manual labor or CNC deburring. CNC machines, which are mainly specialized for milling, are oversized and costly, making them inefficient for the deburring process. The short deburring toolpath also requires frequent workpiece changes, which are best handled by a robotic machine service. The ability of a robot to manipulate the workpiece from an input buffer of a deburring cell, perform the deburring operation and transport the finished product to the output buffer suggests that a robotic deburring solution could be very efficient.

For robotic deburring to be competitive, standard deburring accuracies must be achieved. Compared to CNC machines, robots have some significant disadvantages. In general, inaccuracies are caused by loading conditions that the robot cannot compensate

for. Characteristic sources of inaccuracies in robotic machining include a robot control loop that is not optimized for the machining tasks, backlash hysteresis [2,3] and robot structural and posture effects. Some combinations of robot posture and cutting parameters cause chatter, which can lead to poor machining accuracy or even tool breakage [4–6].

It is interesting to note that the uncertainty and imperfections of a real electromechanical system do not always result in poor system performance, as in some cases it has been found that chaotic control can compensate for mechanical imperfections, which can have a positive effect on the overall performance of the system [7].

To increase the accuracy of robotic machining, many authors have addressed these problems by optimizing the robot trajectory based on a dynamic robot model and the model of the machining process [8–10], as well as by online force control [11,12]. Laser trackers [13] and optical measurements [14] have been used as an alternative to force control to efficiently track the desired path with the robot and achieve accurate machining.

A particular problem in deburring is also the high variance of burr sizes, which can lead to a sudden increase in cutting force, damaging the workpiece, the deburring tool or even the robot. In deburring, the normal force, which is perpendicular to the workpiece surface, and the tangential force, which is aligned with the feed direction, are important. The most common approach to deal with the stochastic forces is to use adaptive algorithms that control the contact force based on online force measurements [15]. The control of tangential force has been successfully solved by reducing the feed rate of the tool [16] and the control of normal force by position correction [17]. An impedance controller has been found to be most effective in achieving a stable contact force between the deburring tool and the workpiece.

However, the implementation of sensors and complex control algorithms in an "out-of-the-box" robotic deburring cell is a major obstacle in transitioning the technology from a clean research laboratory environment to a highly stressed industrial environment. As an alternative, an industry-oriented engineering approach to optimize the deburring parameters considering spindle speed, feed rate and contact force was presented in [18]. In another study, improved control of the robotic deburring process was achieved by shifting the force control from the robot to a simpler external mechanism that is easier to control and designed to compensate for high nominal forces by varying its stiffness [19]. To completely remove larger burrs, several deburring passes were added. The adjustable stiffness of such a mechanism is easy to determine, but the approach still requires some design effort and reduces the flexibility of an autonomous robotic solution. To some extent, force control can also be shifted directly to the cutting tool [20]. To highlight the advantages of robotic deburring, an optimization of the deburring toolpath with respect to the robot stiffness is required. This is essentially a well-known optimization problem for workpiece or trajectory placement [21–23], where robot stiffness is used as the optimization objective [24–28].

To optimize the stiffness of the robot along a trajectory, the stiffness of the robot throughout the workspace must be known. The following two main methods have evolved to determine the stiffness of the robot: global and local [29]. The advantage of the global approach is the ability to capture nonlinear effects due to friction and transmission losses, while the disadvantage is that the model does not generalize well to the entire workspace of the robot. As a result, the established analytical models are sensitive to measurement noise and generate significant errors [30,31]. On the other hand, the more accurate local approach requires the identification of the stiffness matrices of all the robot components, which means that the robot must be disassembled [32]. The scope of analytical modeling of robot stiffness is presented in [33], where a comprehensive six DOF robot stiffness model with 258 identifiable parameters is presented.

Recently, artificial neural networks (ANN) have been discovered in robotics as an alternative to analytical modeling. Their main advantage is that they can approximate nonlinear behavior (which is common in robotics) with the desired accuracy [34,35]. Similar to analytical modeling, ANN modeling requires a large amount of measured data, but in the case of describing the structural properties of a robot, such as stiffness, an ANN

model is much easier to implement and can even improve as new data is collected [36,37]. Since stiffness correlates with accuracy, the kinematic properties of a robot can be used as a general guide for robotic machining [38]. Serial robots, which have the greatest stiffness at the edges of the workspace, are more suitable for machining smaller workpieces, while quasi-serial robots, which have the greatest stiffness in the center of the workspace, are recommended for machining larger workpieces.

A peculiarity of the deburring process is that most of the workpieces in the deburring phase were preproduced on a different setup. Due to the new setup, any uncertainty in the placement of the workpiece can lead to a collision during operation. Therefore, a reference system must be used to detect the position and orientation of the workpiece and adjust the deburring toolpath to the new workpiece offset. As a solution, an iterative nearest point based contour matching algorithm was applied in [39] to match the toolpath extracted from the CAD model to the teach points. Another advantage of autonomous robotic deburring is that the robot can autonomously place the workpiece in the deburring fixture and then perform the required calibration. It has been shown that a robotic machining setup with a static worktable can increase the machining quality, and a robotic machining cell with a rotary table has also been developed [40], but using a rotary table to support the robot to increase the machining performance during operation is new.

The objective of this paper is to study the performance of robotic deburring with a rotary table. The stiffness of the robot is used as a performance measure and modeled using ANN based on measured data. First, the stiffness-based optimization of the workpiece placement is performed, followed by the optimization of toolpath tracking. For toolpath tracking, two scenarios are considered; during cutting motions, the table rotates to position the workpiece such that the robot maintains high stiffness, and during non-cutting motions, the workpiece is repositioned by large rotations according to the stiffness of the robot. The approach is designed to be industry oriented. It integrates components that are common in the machining industry and does not require sensitive sensors or complex algorithms once it is set up.

The paper is organized as follows: In Section 2, the kinematic model of the deburring cell is established. In Section 3, the stiffness model is presented. Section 4 presents the optimization algorithm. Section 5 presents the results. Section 6 concludes the article.

## 2. Robotic Deburring Cell

One possible setup of the robotic deburring cell is shown in Figure 1. The setup shown includes a quasi-serial robot with a deburring head, a rotary table and the workpiece. In this theoretical study, no fixture is considered, as these are usually custom made. In the non-optimized setup, the workpiece is placed in the center of the rotary table.

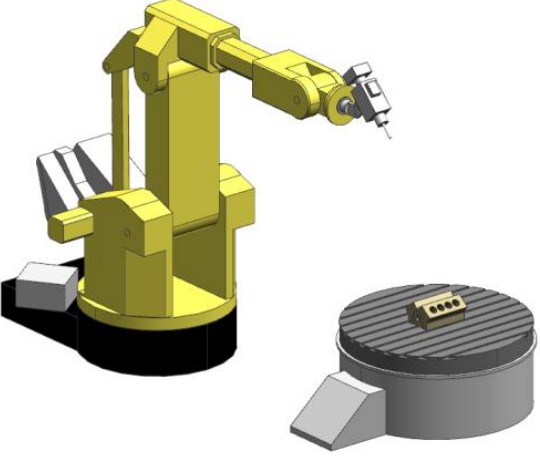

**Figure 1.** Default robotic deburring cell setup.

### 2.1. Kinematic Model of Robot ACMA XR701

The kinematic model of the quasi-serial ACMA XR701 robot is constructed using DH-notation and direct homogeneous transformations. First, a simplified model is constructed in DH-notation by considering only the serial kinematic chain and excluding the kinematic parallelogram. Then, the kinematic parallelogram is added to the simplified model using direct homogeneous transformation matrices (HTM) to construct a detailed kinematic model of the robot (Figure 2).

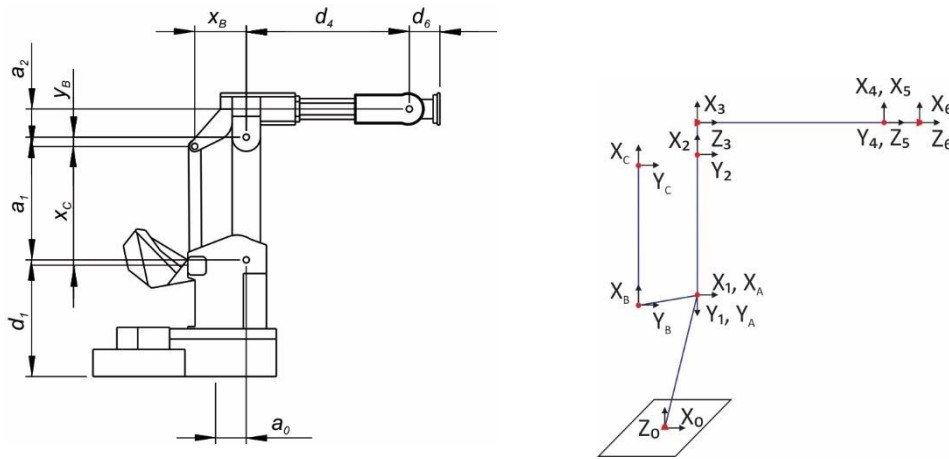

**Figure 2.** Detailed kinematic model of ACMA XR701.

The simplified model is sufficient to describe the transformation from the robot base frame to the end-effector frame required for inverse kinematics. Moreover, solving the inverse kinematics using the simplified model with fewer components is computationally more efficient than using the detailed kinematic model of the robot. Therefore, the simplified model is used for computationally intensive optimization, while the detailed model is used for result validation and presentation.

The DH-parameters for the simplified robot model are shown in Table 1. The simplified kinematic modeling approach considers $\theta_3$ as the actuated joint, which is also intuitively the resulting angle between link two and three. The homogeneous transformation matrix $^0_6T$ between frames zero and six is expressed as follows:

$$^0_6T = {^0_1}T{^1_2}T{^2_3}T{^3_4}T{^4_5}T{^5_6}T \tag{1}$$

**Table 1.** DH-kinematic parameters.

| Link $i$ | $a_{i-1}$ (m) | $\alpha_{i-1}$ (rad) | $d_i$ (m) | $\theta_i$ (rad) |
|:---:|:---:|:---:|:---:|:---:|
| 1 | 0.2500 | $-\pi/2$ | 0.9500 | 0 |
| 2 | 1.0000 | 0 | 0 | $-\pi/2$ |
| 3 | 0.2300 | $-\pi/2$ | 0 | 0 |
| 4 | 0 | $\pi/2$ | 1.3294 | 0 |
| 5 | 0 | $-\pi/2$ | 0 | 0 |
| 6 | 0 | 0 | 0.2500 | 0 |

The detailed kinematic modeling approach with the kinematic parallelogram considers $\theta_A$ as the actuated joint, which is consistent with the kinematics of the real robot. The parallelogram structure is difficult to account for using the DH-approach. A much more convenient approach is to use HTMs to account for the parallelogram. In this way, frames *A*, *B* and *C* are added as an open kinematic chain to frame two of the simplified kinematic model to represent the kinematic parallelogram. The HTM-kinematic parameters for the detailed kinematic model are shown in Table 2.

**Table 2.** HTM-kinematic parameters.

| Link $i$ | $x$ (m) | $y$ (m) | $z$ (m) | $\alpha$ (rad) | $\beta$ (rad) | $\gamma$ (rad) |
|:---:|:---:|:---:|:---:|:---:|:---:|:---:|
| A | 0 | 0 | 0 | 0 | 0 | 0 |
| B | −0.42 | 0.075 | 0 | −$\pi/2$ | 0 | 0 |
| C | 1 | 0 | 0 | 0 | 0 | 0 |

Each individual homogeneous transformation matrix $^{i-1}_{i}T$ is obtained by multiplying the homogeneous transformation matrix $^{i-1}_{i}T_t$ represented by the translation vector $t_i = [x_i, y_i, z_i]$ and the homogeneous transformation matrix $^{i-1}_{i}T_r$ represented by the Euler angles $r_i = [\alpha_i, \beta_i, \gamma_i]$, as expressed by the following:

$$^{i-1}_{i}T = {}^{i-1}_{i}T_t {}^{i-1}_{i}T_r \tag{2}$$

The homogeneous transformation matrix $^{0}_{C}T$ between frames 0 and C can now be expressed as follows:

$$^{0}_{C}T = {}^{0}_{1}T {}^{1}_{2}T {}^{2}_{A}T {}^{A}_{B}T {}^{B}_{C}T \tag{3}$$

where the functional dependence of $\theta_3$, $\theta_B$, $\theta_C$ on $\theta_2$ and $\theta_A$ is expressed as follows:

$$\begin{cases} \theta_3 = -(\theta_2 + \pi/2) + \theta_A \\ \theta_B = (\theta_2 + \pi/2) - \theta_A \\ \theta_C = \theta_A \end{cases} \tag{4}$$

### 2.2. Kinematic Model of the Deburring Head

The deburring head significantly changes the kinematics of the robot. The robot's reach is increased, but at the same time, some areas become more difficult to reach, especially when collisions are considered. The deburring head also influences the structural properties of the robot and, thus, the machining accuracy [41]. Considering both aspects, a deburring head design, as shown in Figure 3, was chosen.

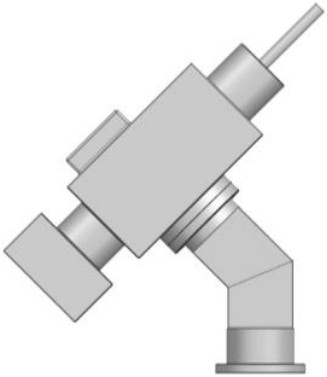
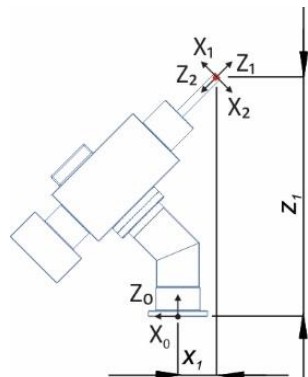

**Figure 3.** Kinematic model of the deburring head.

To include a deburring head in the robot kinematic model, HTMs representing the translation and orientation of the spindle frame $^{2}_{0}T^{tool}$ were multiplied by $^{0}_{6}T^{robot}$. $^{1}_{0}T^{tool}$ is obtained from $t^{tool}_{1} = [-0.075, 0, 0.456]$, which is the translation vector representing the position and $r^{tool}_{1} = [0, -\pi/4, 0]$, which are the Euler angles representing the orientation of the spindle frame relative to the robot end-effector frame $_{6}T^{robot}$. $^{2}_{1}T^{tool}$ is obtained by adding a final rotation by Euler angles $r^{tool}_{2} = [0, 0, \pi]$ to align the orientation of the spindle frame according to the post-processed toolpath. All the position coordinates are in meters and all the orientation coordinates are in radians. The spindle rotation is unlimited.

### 2.3. Kinematic Model of the Rotary Table

Optimal placement of the workpiece allows postures with maximum stiffness to be used in robotic machining. In a static rotary table position, the drilling accuracy could be increased [42]. To exploit this finding, the current research focuses on optimizing the table rotation during deburring.

The kinematic model of the one DOF rotary table is assembled from the following two components: the fixed table base and the rotary clamping plate (Figure 4). The translation vector $t_1^{table} = [1.815, 0.750, 0]$ represents the position of the table base ${}_0T^{table}$ relative to the robot base ${}_0T^{robot}$. The translation vector $t_2^{table} = [0, 0, 0.550]$ represents the position of the rotating clamping plate frame relative to the table base frame and forms the final homogeneous transformation matrix ${}_0^2T^{table}$. The rotation of the rotary table is limited to the interval $rot_{PT,max} \in [-2\pi, 2\pi]$.

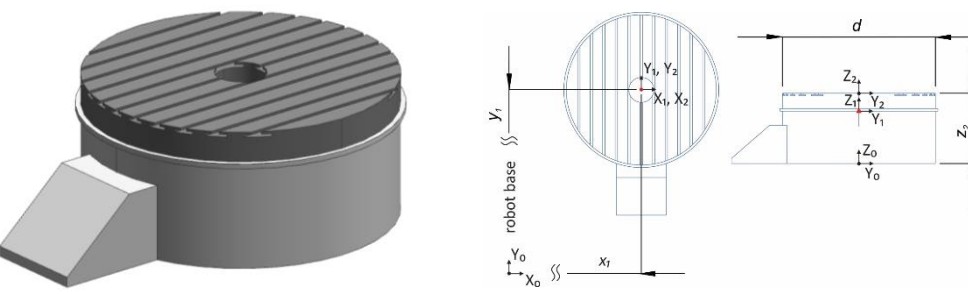

**Figure 4.** Kinematic model of the rotary table.

### 2.4. Workpiece Offset

The workpiece offset $W$ defines the position and orientation of the workpiece in the deburring cell. In our case, an engine block is used as the workpiece and $W$ is presented with the XM-YM-ZM frame (Figure 5). The translation vector $t^W = [x_W, y_W, z_W]$ represents the position of the workpiece offset and the Euler angles $r^W = [\alpha_W, \beta_W, \gamma_W]$ represent the orientation of the workpiece offset relative to the rotating clamping plate frame ${}_0^2T^{table}$. By default, the workpiece offset matrix ${}_0^1T^W$ is identical to ${}_0^2T^{table}$.

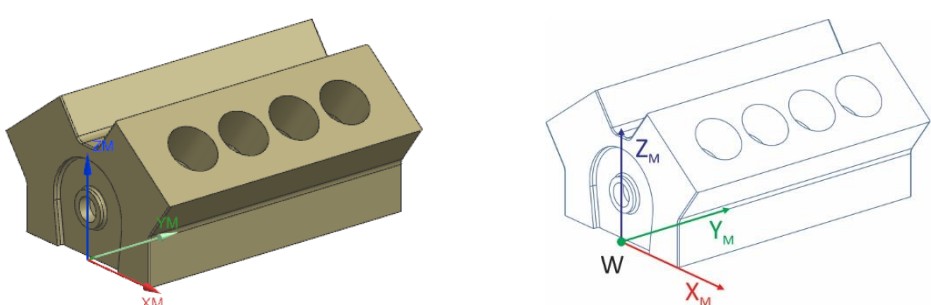

**Figure 5.** Kinematic model of the engine block.

### 2.5. Deburring Toolpath

The deburring toolpath is generated relative to the workpiece offset (Figure 6). In robotic machining, the generated toolpath cannot contain circular interpolations; therefore, all the circular motions are postprocessed as incremental linear motions. In general, the toolpath consists of cutting and non-cutting motions defined between Cartesian points. All the movements must be executed at a specific feed rate and tool orientation to meet the technological requirements. Therefore, each toolpath segment is represented by a frame with an origin defined by three position and three orientation coordinates. With respect to the workpiece offset, the position and orientation of each toolpath segment can be expressed by a translation vector $t^t = [x_t, y_t, z_t]$ and by three Euler angles $r^t = [\alpha_t, \beta_t, \gamma_t]$, where $t = 1, \dots, n$, and $n$ is the total number of toolpath segments.

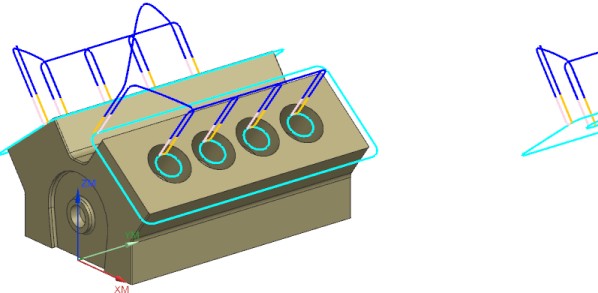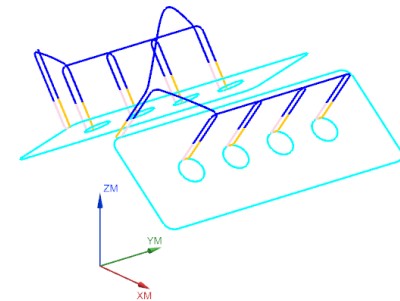

**Figure 6.** Cutting and non-cutting motions on the deburring toolpath.

## 3. Stiffness Model

An approach based on structural stiffness measurements and FEM model calibration was used to build a stiffness model of the ACMA XR701 robot. The full parameter identification procedure is explained in our previous work [43]. The experimental matrix can be found in Table S1.

ANN *stif_net* was used to correlate joint rotations $\theta_2$, $\theta_3$, $\theta_4, \theta_5$ and the load with structural stiffness. To train the ANN *stif_net*, the Levenberg–Marquardt backpropagation algorithm was used with two hidden layers, the first layer with ten neurons and the second with five neurons. The best architecture of the network was found by a random search. To build the structural stiffness model, 70% of the data were used for training, 15% for validation and 15% for testing. Based on the available data, the ANN *stif_net* model errors were found to be between $-54.5793$ and $59.5260$ N/mm, corresponding to an absolute deformation error of $-0.01929$ and $0.046024$ mm, respectively.

The structural stiffness was determined to be load dependent, and the dependence is nonlinear. In the present study, only the output of the structural stiffness model obtained for the end-effector load of 150 N is used. The low load case is considered because the contact forces during deburring are small and mainly depend on the workpiece material and the current burr size. The same load was also used as a limit value for robotic deburring with the support of a variable stiffness mechanism [19]. However, the stiffness modeling approach presented in this work allows lower, even adjustable, stiffness values. With online force measurements, the model can also adapt to the current load. A drawback of the presented approach is the inability of the model to predict stiffness in more than one direction. To be able to align the stiffness ellipsoid of the robot along the direction of the forces applied by the machining process, additional stiffness measurements should be performed on the robot.

Figure 7 shows the structural stiffness $T$ in a section of the theoretical workspace of the robot for the end-effector load of 150 N. The figures were obtained by discretizing the workspace of the robot. Each point represents the end-effector position of the robot without the tool. The color of the point represents the structural stiffness of the robot. The points were generated with rotations of each robot axis as shown in Table 3. In the workspace of the robot, the distribution of the structural stiffness shows two divided areas of low stiffness, with the highest stiffness in the middle of the workspace. This is due to the topology of the robot and is characteristic of quasi-serial robots [44,45].

**Table 3.** Limit values and step sizes for discrete rotations of the robot axis.

|  | $\theta_1$ (°) | $\theta_2$ (°) | $\theta_3$ (°) | $\theta_4$ (°) | $\theta_5$ (°) | $\theta_6$ (°) |
|---|---|---|---|---|---|---|
| min. | 0 | $-50$ | 30 | $-200$ | $-120$ | 0 |
| max. | 0 | 65 | 155 | 200 | 120 | 0 |
| step | 0 | 5 | 5 | 50 | 30 | 0 |

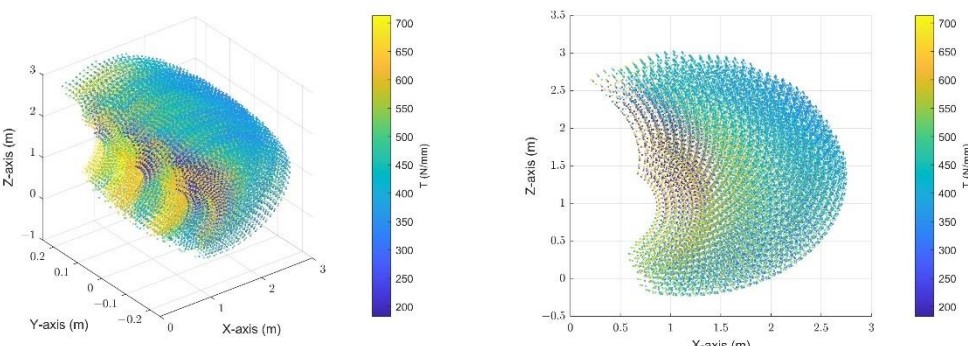

**Figure 7.** Structural stiffness in the Cartesian workspace of the ACMA XR701 robot.

## 4. Optimization of Robotic Deburring

The robotic deburring optimization is divided into two steps. In the first step, the GA is used. The objective is to find the optimal initial position of the workpiece that allows the robot to debur the workpiece with the maximum total stiffness. The total structural stiffness of the robot along the entire toolpath is considered as an objective function and evaluated when the entire toolpath has been traversed from the current initial position. If the robot is unable to track the toolpath or violates predefined constraints, the objective function is penalized. The first optimization step is described in more detail in Section 4.1.

In the second optimization step, a one-dimensional minimizer is used. The objective is to increase the stiffness of the robot while it tracks the deburring toolpath. The rotary table is used to help the robot to adopt a posture with high stiffness on each successive toolpath segment. The structural stiffness of the robot is considered as an objective function. The second optimization step is described in more detail in Section 4.2.

### 4.1. Workpiece Initial Placement

To find the initial position and orientation of the workpiece that allows the robot to debur the workpiece with the maximum overall stiffness, an optimization was created using a genetic algorithm (GA). The general GA procedure is shown in Table 4, where $S$ represents the variables, $S^*$ the search space, $P$ the population size, $E$ the constrained space and $C$ the objective function.

**Table 4.** Pseudocode of the genetic algorithm.

| | |
|---|---|
| 1 | Initialize $P_t$ generation with random individuals from $S^*$ |
| 2 | For each individual $i \in P_t$ |
| 2.1 | Evaluate $C(S, P_t)$ |
| 3 | Until convergence, do |
| 3.1 | Reproduction of best individuals from $P_t$ (based on $C$) |
| 3.2 | Crossover of random individuals from $P_t$ |
| 3.3 | Mutation of random individuals from $P_t$ |
| 3.4 | $P_{t+1} \leftarrow$ create new generation |
| 3.5 | For each individual $i \in P_{t+1}$ |
| 3.5.1 | Evaluate $E(S, P_{t+1})$ |
| 3.5.2 | For each individual $i \notin E$ |
| 3.5.2.1 | Evaluate $C(S, P_{t+1})$ |
| 4 | Retrun best individual from $P$ |

#### 4.1.1. Variables, Search Space and Population Size

The placement of the workpiece offset is a six DOF problem. To optimize the placement of the workpiece offset, which determines the placement of the workpiece, we define three position $[x_{abs}, y_{abs}, z_{abs}]$ and three orientation parameters $[\alpha_{abs}, \beta_{abs}, \gamma_{tabs}]$. In practice, the placement of the workpiece is constrained by the deburring setup. In the algorithm, the constraints are considered by the search space (Table 5). In our setup, to find the

optimal placement of the workpiece on the rotary table, only three variables are considered $x_{var}$, $y_{var}$, $\alpha_{var}$. Considering the placement of the workpiece in the variable Z-direction or with variable rotation around the *X*- or *Y*-axis is technically challenging as it requires a very complex fixture to hold the workpiece. To allow variable placement of the workpiece with rotation around the *X*- or *Y*-axis, the one DOF rotary table should rather be replaced by a two DOF rotary table.

**Table 5.** Search space defined by lower and upper bounds of the variables.

|     | $x_{var}$ (m) | $y_{var}$ (m) | $z_{var}$ (m) | $\alpha_{var}$ (rad) | $\beta_{var}$ (rad) | $\gamma_{var}$ (rad) |
|-----|------|------|------|------|------|------|
| LB  | 1.255 | 0.190 | 0 | $-\pi$ | 0 | 0 |
| UB  | 2.375 | 1.310 | 0 | $\pi$ | 0 | 0 |

The position and orientation parameters $z_{abs}$, $\beta_{abs}$, $\gamma_{abs}$ are preset as the following:

$$\begin{cases} z_{abs} = 0 \\ \beta_{abs} = 0 \\ \gamma_{abs} = 0 \end{cases} \tag{5}$$

The remaining position and orientation parameters $x_{abs}$, $y_{abs}$, $\alpha_{abs}$ are expressed as the following:

$$\begin{cases} x_{abs} = x_{var} - 1.815 \\ y_{abs} = y_{var} - 0.750 \\ \alpha_{abs} = \alpha_{var} \end{cases} \tag{6}$$

where $x_{var}$, $y_{var}$, $\alpha_{var}$ are the optimization variables. The population size was set to 50.

#### 4.1.2. Nonlinear Constraints

Two nonlinear constraints are considered in the algorithm to avoid undesired solutions. The first nonlinear constraint restricts the combinations of the variables $x_{var}$ and $y_{var}$ to the area within the cylinder defined by the outer edge of the rotary table as follows:

$$(x_{var} - x_0)^2 + (y_{var} - y_0)^2 < r^2 \tag{7}$$

where $x_0, y_0, r = d/2$ are the rotary table kinematic parameters (Figure 4):

$$\begin{cases} x_0 = 1.815 \\ y_0 = 0.750 \\ r = 0.560 \end{cases} \tag{8}$$

Considering $z_{abs} = 0$, this means that the algorithm places the workpiece offset *W* exclusively on the rotary table.

The second nonlinear constraint restricts the orientation of the workpiece offset *W* to one full revolution, as follows:

$$\alpha_{var}^2 \leq pi^2 \tag{9}$$

This ensures that the cyclically equivalent placements of the workpiece are not considered by the algorithm.

#### 4.1.3. GA Objective Function

The algorithm must ensure that the robot performs the deburring task with the highest total structural stiffness. Since the optimization is a minimization, the inverse of the stiffness at each toolpath segment is evaluated.

The inverse of the robot's stiffness is obtained by simulating ANN *stif_net* with the current robot configuration under the current external load (*cur_config_load*), as follows:

$$T^{inv} = (sim(stif\_net, cur\_config\_load))^{-1} \tag{10}$$

For this theoretical study, an external load of 150 N was used throughout the toolpath. To maximize the total structural stiffness over the entire deburring path, the objective function can be expressed as follows:

$$T_{tot}^{inv} = \sum_{i=1}^{n} T_i^{inv}$$

(11)

where $n$ is the total number of toolpath segments.

### 4.1.4. Objective Function Constraints

Some workpiece offsets accepted by the algorithm may result in an undesirable toolpath placement that makes it impossible to debur the workpiece with the desired accuracy. The individuals that lead to an undesirable toolpath placement are artificially penalized to degrade their fitness and reduce their chances of advancing to the next generation. The traits of the punished individuals will be less common in the individuals that form the future generations.

The first case of undesirable toolpath placement is when at least one toolpath segment lies outside the cylinder defined by the outer edge of the rotary table, as follows:

$$(x_t - x_0)^2 + (y_t - y_0)^2 < r^2, \ t = 1, \dots, n$$

(12)

where $x_t$ and $y_t$ are the coordinates of the current machining toolpath segment and $n$ is the total number of toolpath segments. At such a machining toolpath segment, a workpiece protrusion occurs, which reduces the clamping stiffness of the workpiece. The clamping stiffness of the workpiece is considered negligible without a protrusion.

Another case of undesirable toolpath placement is when a toolpath segment cannot be reached due to the robot's axis limits. The kinematics of the robot are different from that of a machine tool, which usually has at least three linear axes; therefore, some seemingly easy-to-reach toolpath segments are unreachable by the robot. In addition, the robot is equipped with a deburring head, which further increases the kinematic complexity of the system.

The limit values for the position axes with respect to the home position of the robot, as defined by the DH-parameter values in Table 1, are as follows:

$$\begin{cases} \theta_1 \in (-180°, 180°) \\ \theta_2 \in (-55°, 65°) \\ \theta_3 \in (-60°, 65°) \end{cases}$$

(13)

where $\theta_1$, $\theta_2$, $\theta_3$ are the first three axes of the robot.

The limit values for the orientation axes are as follows:

$$\begin{cases} \theta_4 \in (-200°, 200°) \\ \theta_5 \in (-120°, 120°) \\ \theta_6 \in (-300°, 300°) \end{cases}$$

(14)

where $\theta_4$, $\theta_5$, $\theta_6$ are the last three axes of the robot.

The last considered case of undesired toolpath placement is when at least one toolpath segment cannot be reached by the robot with the desired accuracy. For position coordinates, the restriction is set above the standard CNC machining accuracy of $3 \cdot 10^{-5}$ m. The acceptable deburring accuracy is defined as follows:

$$\begin{cases} |x_i - x_{i,des}| < 3 \cdot 10^{-6} \\ |y_i - y_{i,des}| < 3 \cdot 10^{-6} \\ |z_i - z_{i,des}| < 3 \cdot 10^{-6} \end{cases}$$

(15)

For orientation, the restriction is set as follows:

$$
\begin{cases}
\left| \alpha_i - \alpha_{i,des} \right| < 3 \cdot 10^{-4} \\
\left| \beta_i - \beta_{i,des} \right| < 3 \cdot 10^{-4} \\
\left| \gamma_i - \gamma_{i,des} \right| < 3 \cdot 10^{-4}
\end{cases}
\tag{16}
$$

### 4.2. Tracking the Toolpath

The second optimization step is to ensure that the robot tracks the deburring path with the highest possible overall structural stiffness, starting from an initial workpiece placement determined by the GA. A rotary table is used to position the workpiece during deburring to assist the robot in maximizing its stiffness. One table rotation is allowed for each successive segment of the toolpath. For each rotation, the workpiece is repositioned, and the robot must adopt a new posture to reach it. For cutting motions, a limited rotation of the rotary table is allowed to avoid collisions due to the relative motion between the tool and the workpiece. For non-cutting motions, the rotary table is used to reposition the workpiece with unrestricted rotation.

The optimization problem to determine the stiffest robot posture for the next segment of the toolpath is formulated as a one-dimensional maximization problem with fixed constraints. The inverse of the structural stiffness of the robot, as given in Equation (10), was used as the objective function.

For computational efficiency, every tenth segment of the toolpath was considered in the optimization; therefore, the optimized toolpath consisted of 155 segments. The optimization problem was solved using the MATLAB function *fminbnd*.

### 4.2.1. Rotary Table Constraints

The deburring toolpath consists of cutting and non-cutting motions. During non-cutting motions, the tool is not in contact with the workpiece. The tool moves to a clearance where no collisions are possible, and a large rotation of the rotary table can be allowed. During non-cutting motions, unrestricted table rotation is permitted, in the following interval:

$$
rot_{t,n} \in (-\pi, \pi)
\tag{17}
$$

During cutting motions, the tool is in contact with the workpiece, which means that excessive table rotation can cause damage to the workpiece. To keep the inaccuracies caused by the relative motion of the workpiece and the tool within an acceptable range, the maximum allowable table rotation is set based on the deburring accuracy tolerance. Depending on the current toolpath segment $c_t = [x_t, y_t, z_t]$, the relative motion between the tool and the workpiece can be expressed as the Euclidean distance between two points, as follows:

$$
dist = \sqrt{(x_{t+1} - x_t)^2 + (y_{t+t} - y_t)^2 + (z_{t+1} - z_t)^2}
\tag{18}
$$

where

$$
c_{t+1}(x,y,z) = r_z \cdot c_t(x,y,z)
\tag{19}
$$

and $r_z$ is the counterclockwise rotation around the $Z$-axis.

For an acceptable deburring accuracy of less than $3 \cdot 10^{-5}$ m, the maximum rotation of the rotary table at each toolpath segment must be in the following interval:

$$
rot_{t,c} \in \left( -\pi \cdot 10^{-5}, \pi \cdot 10^{-5} \right)
\tag{20}
$$

If the current toolpath segment is at the outer edge of the table, the maximum relative motion between the workpiece and the tool is $2.4880 \cdot 10^{-5}$ m.

### 4.2.2. Current Table Configuration

To obtain the rotation of the rotary table that allows the robot to operate with the highest stiffness, a one-dimensional minimizer is used. The initial table configuration is set

to $rot_{t,c} = 0$, while for each subsequent toolpath segment, the new configuration is found by searching for the maximum stiffness of the robot on the interval $rot_{t,c}$.

## 5. Results and Discussion

The first solution was obtained as an inverse kinematic problem with the engine block placed in the center of the table, as shown in Figure 1. With the non-optimized setup, the robot manages to debur the outer edges on one side of the workpiece but exceeds the joint limits before reaching the cylinder bores, even though the table rotation is enabled, as explained in Section 4.2.

Several feasible solutions were found by optimizing the workpiece placement. The best solution was found with the minimum total inverse of stiffness along the toolpath at 0.339 mm/N. The corresponding optimization variables are $x_{var} = 1.631988$ m, $y_{var} = 1.075345$ m and $\alpha_{var} = 2.6278$ rad. The optimized setup is shown in Figure 8.

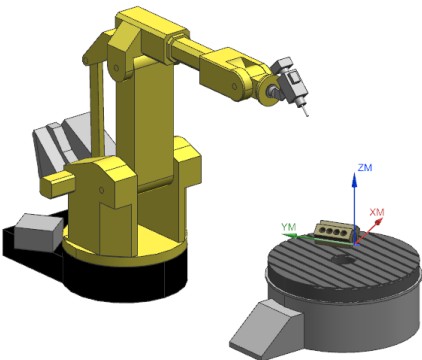

**Figure 8.** Optimized robotic deburring cell setup.

Toolpath tracking was divided into the following two types of motion: cutting motion and non-cutting motion. Based on the type of motion in the current toolpath segment, different sizes of rotary table rotation were allowed. Table 6 shows the maximum rotational difference on the representative toolpath segments when deburring or moving between the deburring features.

**Table 6.** Maximum rotational difference on different toolpath segments.

|  | **Outer Edges** | **Bore 1 Edges** | **Bore 1–4** | **Traverse** |
|---|---|---|---|---|
| Table rotation (°) | 0.12 | 0.05 | 5.07 | 147.50 |

A comparison between the original (a) and the optimized (b) toolpaths is shown in Figure 9. Both toolpaths are shown relative to the base frame of the robot. The optimized toolpath spans a larger workspace and gives the robot more freedom to adopt a stiff posture. The largest distortions seen in the Cartesian representation of the optimized deburring toolpath occur during non-cutting motions. This can be observed even for short segments such as the path between the deburring of cylinder holes, where the toolpath is compressed into a smaller space compared to the original toolpath. Toolpath distortion is virtually non-existent during cutting motions because the permissible table rotations and the number of toolpath segments are small.

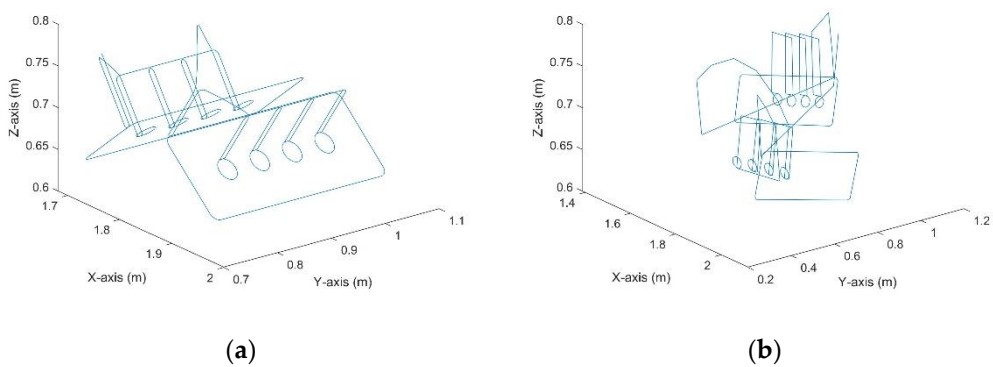

**Figure 9.** The original (**a**) and the optimized (**b**) deburring toolpath.

When deburring the outer edges on the left side of the engine block, the maximum rotational difference of the table is 0.12°. It should be noted that the toolpath segments created for deburring the outer edges sometimes require long linear motions. To reach the next toolpath segment during a long linear motion, the robot must travel a long distance, which can lead to a reduction in stiffness or even a constraint violation. To increase the performance of the robot and reduce the risk of constraint violation, the table rotation could be activated during the long linear motions. This would require breaking down the long linear motions into smaller toolpath segments, which was not performed in this study.

When deburring the edges of the first cylinder bore, the maximum rotational difference of the table is 0.05°. One reason for the small rotational difference is that only every tenth toolpath segment was considered in the optimization. If the complete toolpath was considered, each additional segment would also contribute to the maximum rotation difference. Second, the absolute maximum table rotation was evaluated for a worst-case scenario. If the Euclidean distance, based on which the absolute maximum table rotation was calculated, were to take the current toolpath segment into account, a larger maximum table rotation difference could also be achieved. Between the deburring of each edge and each cylinder bore, non-cutting motions occur. During the non-cutting motions, free table rotation is enabled. The table rotation during deburring of the first and the fourth cylinder bore on the left side of the engine block was 5.07°. Despite the possibility of a larger table rotation, only a small rotation is performed to reposition the workpiece to the area with the highest stiffness of the robot, which shows that the workpiece placement found by the optimization algorithm is robust. Figure 10 shows the posture of the robot and the rotation of the table when deburring the first cylinder bore on the left and when deburring the fourth cylinder bore on the right.

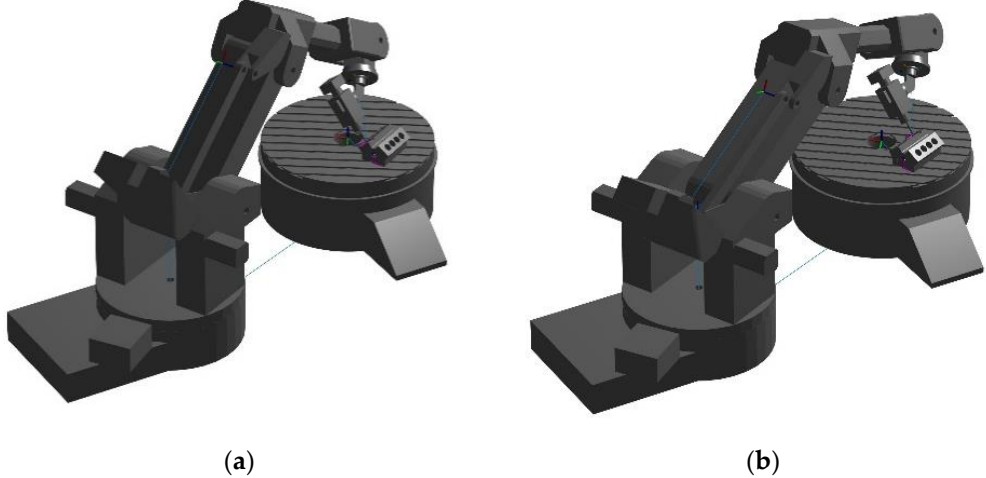

**Figure 10.** Optimized deburring of (**a**) cylinder bore 1 and (**b**) cylinder bore 4.

The most problematic segment of the toolpath for the robot is the traverse from the left to the right side of the engine block. Most optimizers have failed on this motion. In the optimized case, a table rotation of 147.50° was performed. Figure 11 shows the posture of the robot and the rotation of the table when deburring is changed from the left side of the engine block, as shown on the left, to the right side of the engine block, as shown on the right. Due to the large change in posture of the robot, it is interesting to compare the performance of the robot in deburring each side. The average inverse of stiffness is 0.00213 mm/N when working on the left side of the engine block and 0.00226 mm/N when working on the right side. It is also interesting to note that the absolute maximum rotational difference when working on the left side is 37.82° and when working on the right side is 8.92°, which means that the robot is less flexible when working on the right side. The higher maximum rotational difference of the table when the robot works on the left side of the engine block is mainly due to a large rotation when changing from the outer edge to the edges of the cylinder bores.

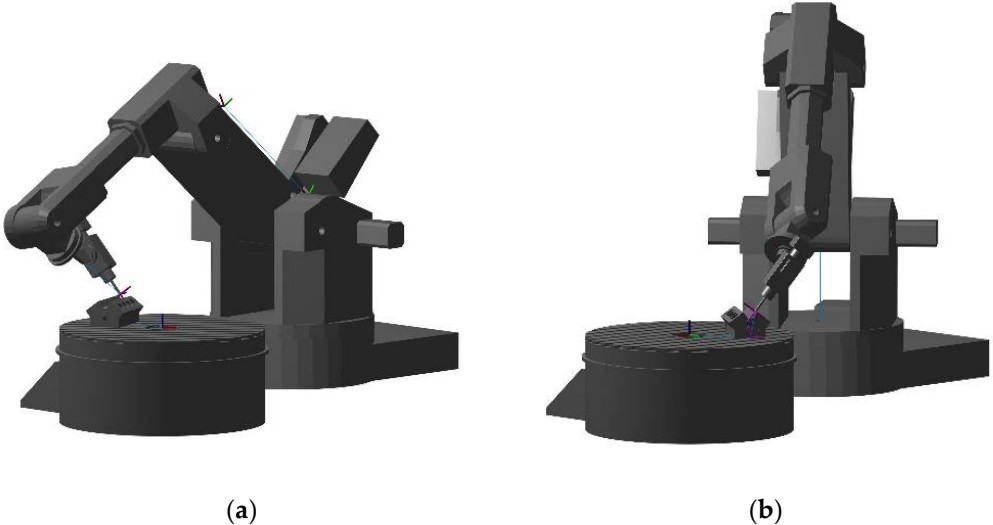

(**a**)                                                       (**b**)

**Figure 11.** Traverse between (**a**) the left side and (**b**) the right side of the engine block.

## 6. Conclusions

In this study, a methodology for planning a deburring toolpath of a robot considering its structural stiffness is presented. When deburring a large workpiece, such as a cast iron or aluminum alloy engine block, path optimization is advantageous because the robot's stiffness varies greatly over its workspace, which affects the quality of the process. The same is true when the parts produced by machining operations such as grinding, drilling, milling, engraving or turning are intended for robotic deburring. The proposed methodology helps in planning efficient deburring, especially when the workpiece is large and complex in shape.

To improve the deburring performance, an industry-oriented robotic deburring cell with integrated rotary table is presented, which helps the robot to increase the effective reach and maintain high stiffness. To achieve the required deburring accuracy and still exploit the functionality of the rotary table, large rotations of the table are allowed during non-cutting motions and small rotations of the table are allowed during cutting motions. To maximize the overall stiffness of the robot along the deburring toolpath, an optimization of the workpiece placement is first performed, followed by an optimization of the toolpath tracking. A genetic algorithm is used to find the optimal workpiece placement and a local minimizer is used to find the optimal table rotation at each toolpath segment.

The structural stiffness of the robot is modeled by a novel approach using an ANN. The advantage of ANN modeling is the possibility to collect data through global stiffness measurements. In this way, the ANN stiffness model considers nonlinear effects such as friction, wear, and gaps between the components, in addition to the full flexibility of all the

joints and links and allows the structural stiffness to be considered as a function of load. The errors and trends of the ANN structural stiffness model are comparable to the results of other studies on similar robots.

The optimization results show that the deburring of large workpieces, such as an engine block, must be carefully planned. By placing the engine block in the middle of the rotary table, the robot was not able to track the toolpath. With the optimized setup, the robot can track the toolpath, but since the effective table rotations are small, the deburring performance could be further improved. The main reason for the small table rotations is that table rotations only occur at the beginning of each toolpath segment and only to an extent that does not affect the required deburring accuracy, which was evaluated in a worst-case scenario. Larger table rotations could be enabled by considering the coordinates of the current toolpath segment. In addition, only every tenth toolpath segment was considered in the optimization and the long linear motions were not broken down into smaller segments. If each toolpath segment was considered and the long linear motions were broken into smaller toolpath segments, the table rotations would occur more frequently. However, increasing the number of toolpath segments would not only increase the computational effort, but also the effort for generating the toolpath, since the decomposition of the toolpath into smaller segments is not readily possible in the CAM software.

The optimization also showed that many optimizers did not find a solution, mainly because the robot could not reach both sides of the engine block. Moreover, the optimized solution showed better flexibility and average stiffness of the robot when it worked on one side of the engine block than when it worked on the other side. This imbalance could be reduced by allowing the table to rotate during long linear movements. However, a more general conclusion is that the placement of the table should be optimized before building the cell, or that additional axes such as a rail should be considered for the robot to increase the kinematic freedom of the system. In addition to validating the results experimentally, these concerns will be addressed in our future work.

**Supplementary Materials:** The following are available online at https://www.mdpi.com/article/10.3390/app11178213/s1.

**Author Contributions:** Conceptualization, J.G.; methodology, J.G.; experiments, J.G. and T.K.; software and simulation, J.G.; investigation, J.G. and T.K.; writing—original draft preparation, J.G.; writing—review and editing, J.G. and T.K.; supervision, M.B. All authors have read and agreed to the published version of the manuscript.

**Funding:** The authors thank the Slovenian Ministry of Higher Education, Science and Technology and the Slovenian Research Agency (Research Core Funding No. P2-0157) for financial support that made this work possible. The authors also acknowledge financial support from the ROBKONCEL project (OP20.03530).

**Institutional Review Board Statement:** Not applicable.

**Informed Consent Statement:** Not applicable.

**Data Availability Statement:** Not applicable.

**Conflicts of Interest:** The authors declare no conflict of interest.

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
