# Peer review of "Stiffness-Based Cell Setup Optimization for Robotic Deburring with a Rotary Table"

_applsci, doi:10.3390/app11178213_

Round 1

Reviewer 1 Report

The paper deals with the robotic deburring by introducing a 2 steps optimization: in the first step, through a GA-based algorithm using a cost function that maximizes the robot stiffness, the initial position of the workpiece on the rotary table is found. In the second step, the robot kinematics configuration is optimized along the deburring path again to maximize the robot stiffness. The method exploits the presence of a rotary table to optimize the robot kinematics configuration (thanks to the kinematics redundancy) to improve the robot's overall stiffness. 
The paper addresses a quite common problem for robotic machining that is the machining path optimization. Currently, most commercial CAD-CAM  generates the robot tool path managing the presence of external axes by generating interpolated trajectories. In this case, the optimization criteria are mostly based on kinematics such as the distance from joints limits, trajectory points reachability, etc... so from the industrial state-of-the-art point of view, the only novelty that I can find in the paper is the evaluation of the robot stiffness in the optimization process. Anyway, looking at the scientific literature also the use of robot stiffness was already evaluated to optimize robot trajectories such as in:   

- C. Reinl, M. Friedmann, J. Bauer, M. Pischan, E. Abele, and O. von Stryk. Model-based off-line compensation of path deviation for industrial robots in milling applications. In IEEE/ASME International Conference on Advanced Intelligent Mechatronics (AIM), pages 367–372, July 2011. doi: 10.1109/AIM.2011.6027113.

and in the Ph.D thesis:

- E. Villagrossi. Robot dynamics modeling and control for machining applications. Ph.D. thesis, University of Brescia, 2016.

As additional remarks:

- is the second step optimization also based on GA? In this case, the cost function should also include the model of the machining process to optimize robot stiffness on the base of the forces acting during the process and to generate trajectories able to orient the robot stiffness ellipsoid along the direction of the forces imposed by the machining process. At least the authors should discuss this problem.
- In the second step the limits imposed to the rotary table movements within a single toolpath segment are really narrow, which is equivalent to keep the position of the table fixed. Do the authors evaluated to generate interpolated movements between the robot and the rotary table during the optimization?
- The two steps method should be better explained and detailed, the second step is not clear from the paper explanation.
- The experimental part is really weak. I would expect a comparison between an optimized trajectory and a not optimized one, in particular, the comparison of deburring performance between two deburred pieces. 
- Which kind of deburring will be addressed after trajectory optimization? Based on my experience, in the case of the deburring of ductile materials, such as aluminum alloys, with small burrs derived from the die casting process, this kind of optimization will not be appreciable.

Based on my previous comments this paper needs to be improved to be evaluated for a journal publication.

The English level is appropriate.

Below I reported some works on the same topic that can be used to improve the bibliography.

Zhang H., Wang J., G. Zhang, G. Zhongxue, P. Zengxi, C. Hongliang, and Z. Zhenqi. Machining with flexible manipulator: toward improving robotic machining performance. In Proceedings of IEEE/ASME International Conference on Advanced Intelligent Mechatronic, pages 1127–1132, July 2005. doi: 10.1109/AIM.2005.1511161

E. Abele, J. Bauer, S. Rothenbücher, M. Stelzer, and O. Stryk. Prediction of the tool displacement by coupled models of the compliant industrial robot and the milling process. In Proceedings of the International Conference on Process Machine Interaction, pages 223–230, Sept 2008.

C. Lehmann, M. Halbauer, D. Euhus, D. Overbeck. Milling with industrial robots: Strategies to reduce and compensate process force induced accuracy influences. Proceedings of 2012 IEEE 17th International Conference on Emerging Technologies Factory Automation (ETFA 2012). doi: 10.1109/ETFA.2012.6489741.

O. Sörnmo, B. Olofsson, A. Robertsson, and R. Johansson. Increasing time-efficiency and accuracy of robotic machining processes using model-based adaptive force control. In 10th IFAC Symposium on Robot Control, pages 543–548. IFAC, 2012. ISBN 978-3-902823-11-3. URL http://dx.doi.org/10.3182/
20120905-3-HR-2030.00065.

Beschi, M., Mutti, S., Nicola, G., Faroni, M., Magnoni, P., Villagrossi, E., Pedrocchi, N. Optimal Robot Motion Planning of Redundant Robots in Machining and Additive Manufacturing Applications. Electronics 2019, 8, 1437. https://doi.org/10.3390/electronics8121437.

And some PhD thesis related to the topic of the paper:

- S. Moberg. On modeling and control of flexible manipulators. Licentiate Ph.D. thesis, University of Linköping, 2007.

- A. Dalla Libera. Learning Algorithms for Robotics Systems. Ph.D. thesis, University of Padova, 2019.

Reviewer 2 Report

This is a well-written, timely and interesting article. I recommend making the following adjustments to the article:
- on page 2, line 70 repeats part from previous sentence ("requires some design effort and reduces the flexibility of an autonomous robotic solution.")
- Figure 8 overlaps the bottom of Table 6.
- I recommend moving all tables and figures under the paragraph in which they are first mentioned.

Reviewer 3 Report

  • To complete the deburring a 6-DOF robot arm still has many problems. Giving a trajectory along the edge of the workpiece can achieve deburring processes, but it cannot ensure the contact force between the tool and the workpiece. So, the contact force is more important to precisely control the force applied by the end-effector rather than controlling the robot’s position. The structural stiffness and positioning accuracy of a typical CNC machine is sufficient for precision deburring. However, CNC machine has limited workspace because most of CNC machines are a Cartesian coordinate robot. The advantage of robotic deburring is it can carry out deburring for the large and complex workpiece. But the structural stiffness of a manipulator is very low. Because the contact force between the deburring tool and workpiece significantly decreases the positioning accuracy in the deburring process. To achieve a stable contact force and obtain a smooth surface after deburring, active force control such as impedance control has been introduced. So, how to deal with the vibration suppression signals from this article?

  • The necessary information of the force model is the burr’s height and the cross-section area of the burr. Not only to achieve the high-level automatic control, the desired trajectory is obtained from CAD/CAM software, but also consider the posture of the robot arm on the free-curved surface. So, how do you solve this problem from this article? (This information is obtained by the vision system?)

  • To my knowledge, hybrid positive/force control is suitable for constrained environments. The “hybrid” technique combines force and torque information with positional data to satisfy simultaneous position and force trajectory constraints specified in a convenient task-related coordinate system. In order to eliminate the control chattering, so, how do you solve discontinuous control phenomena?

  • The article proposes a method for optimization for robotic deburring with a rotary table. However, fixing the robot arm’s end-effector cannot be suitable for complicated workpieces. So, how do you solve this problem from this article?

  • The paper does not have any experimental results to validate the numerical simulation results in section 5. Pls provide and make the comparison.

Round 2

Reviewer 3 Report

The technical work appears to be correct.

The reviewer thinks some parts of the paper have new knowledge.

This paper has been modified according to the previous reviewers’ opinions and suggestions.

The additional material has provided substantial clarity and is much easier to follow.

So it has a logical structure and clear expression.

Author Response

Thank you for your valuable comments and your time.

Sincerely,

Janez Gotlih